# Development of a Competitive Enzyme-Linked Immunosorbent Assay Targeting the-p30 Protein for Detection of Antibodies against African Swine Fever Virus

**DOI:** 10.3390/v15010154

**Published:** 2023-01-04

**Authors:** Junming Zhou, Yanxiu Ni, Dandan Wang, Baochao Fan, Xuejiao Zhu, Jinzhu Zhou, Yiyi Hu, Li Li, Bin Li

**Affiliations:** 1Institute of Veterinary Medicine, Jiangsu Academy of Agricultural Sciences, Key Laboratory of Veterinary Biological Engineering and Technology, Ministry of Agriculture, Nanjing 210014, China; 2Jiangsu Key Laboratory for Food Quality and Safety-State Key Laboratory Cultivation Base of Ministry of Science and Technology, Nanjing 210014, China; 3Jiangsu Co-Innovation Center for Prevention and Control of Important Animal Infectious Diseases and Zoonosis, Yangzhou 225009, China

**Keywords:** African swine fever, monoclonal antibodies, epitope, competitive ELISA, p30

## Abstract

African swine fever (ASF) is a highly contagious hemorrhagic viral disease of domestic and wild pigs of all breeds and ages, caused by African swine fever virus (ASFV). Due to the absence of a safe and efficacious vaccine, accurate laboratory diagnosis is critical for the control of ASF prevention. The p30 protein is immunogenic and stimulates a high level of antibody response to ASFV infection. We developed a panel of 4 monoclonal antibodies (mAbs) against p30 protein, and mAb-2B4 showed the highest percent of inhibition (PI) of 70% in the solid phase blocking ELISA (bELISA). Epitope mapping revealed the mAb-2B4 recognized the epitope of aa 12–18 of p30, which is conserved among various ASFV genotypes. Subsequently, a competitive enzyme-linked immunosorbent assay (cELISA) was established using HRP-labeled mAb-2B4. The cutoff for discrimination between 98 negative sera and 40 positive sera against ASFV was determined by plotting a receiver operating characteristic (ROC) curve. It yielded the area under the curve (AUC) of 0.998, and a diagnostic specificity of 97.96% and a sensitivity of 97.5% were achieved when the cutoff value was determined at 37.1%. Furthermore, the results showed an excellent repeatability of the established cELISA and no cross-reaction to antisera against six other pig pathogens. Additionally, the cELISA detected a titer of 1:256 in the positive standard serum. Overall, mAb-2B4 showed a conserved epitope and high ability to be inhibited by positive sera in ASFV antibody detection. The cELISA based on HRP-labeled mAb-2B4 offers an alternative to other assays for a broader diagnostic coverage of ASFV infection.

## 1. Introduction

African swine fever (ASF) is an ancient disease of domestic pigs, first reported in Kenya in 1921 [1]. It is caused by African swine fever virus (ASFV), the sole member of the family *Asfarviridae*, genus *Asfivirus* [2]. Since the virus crossed the Black Sea into Georgia, its transmission seems to have accelerated [3]. Until now, the worldwide distribution of ASF has resulted in serious economic losses and significant social influence [4,5]. In Africa, warthogs are the main reservoir for the ASFV; however, wild boars are regarded as the primary reservoir in Europe [3]. Overall, ASF spreads by either direct contact with the diseased animal or indirect contact with contaminated environmental factors [6]. Currently, there are no medical treatments or vaccines available for ASF. Therefore, it is critical to prevent virus entry via an ongoing surveillance program for the pig in the ASF-free areas. The severity of the diseases largely depends on the virus virulence. Its incubation period in nature is usually 4–19 days, and the acute form is 3–4 days. Sudden death with few signs usually happens in peracute form, while the chronic form could develop over 2–15 months. The mortality rate of ASF varies from 100% in acute form to less than 20% in chronic form [7]. A low number of survivors might become carriers, who shed the virus and initiate new outbreaks. After the appearance of Georgia-07-like genotype II ASFV in China in 2018, it caused acute disease with almost 100% mortality [8]. Natural mutations in virulent viruses caused the emergence of lower virulent genotype II ASFVs in China in 2020 [9]. Furthermore, genotype I ASFVs had emerged in China, which were highly transmissible, causing chronic and persistent infections in pigs [10]. Due to the longer incubation, low-level viremia, and mild manifestation of these low virulent ASFVs, the diagnosis and control of ASF became complicated [10].

PCR-based diagnosis methods detect ASFV with high sensitivity and specificity. Furthermore, the serological testing of antibodies is a crucial method to diagnose ASF. It can also illustrate epidemiological features of outbreaks such as the incubation period of the viral infection and the infections caused by the low pathogenic strains of ASFV [11]. The routine diagnostic method for ASF approved by the World Organization of Animal Health (WOAH) is the ELISA after preliminary screening, followed by Western blotting [12,13]. Either in endemic or ASF-free countries, the ELISA is commonly used to diagnose viral diseases and to assess serosurveys. The European Union has authorized three antibody ELISAs, including Ingenasa, IDvet, and Svanovir, with an average accuracy rating of about 80% [11]. At present, numerous ELISA-based serological tests are based on the structural protein [14,15,16]. It was found that p30 was the best diagnostic antigen among recombinant p30, p72, and p54 by Cubillos et al. [17].

The indirect ELISA format has proven useful for the serodiagnosis of ASF, and competitive or blocking ELISAs are particularly useful when a higher level of specificity is required [18]. There is a correlation between the promotion of specificity and the isotype and target specificity of the monoclonal or polyclonal antibodies. There are many applications for monoclonal antibodies (mAbs), ranging from the study of therapeutic applications to the defining of epitopes. It will make it easier to clarify viral pathogenesis and the immune response of the host. Furthermore, in the serological diagnosis of viral diseases, mAbs are the most widely used antibodies [13]. In this study, we generated four monoclonal antibodies targeting p30, and mapped the epitope of one mAb which showed good ability to distinguish the positive and negative serum of ASF. Hereafter, we developed an efficient cELISA by HRP-labeled mAb.

## 2. Materials and Methods

### 2.1. Expression and Purification of Recombinant p30 in Escherichia coli

The p30/CP204L genome of ASFV Pig/HLJ/2018 (accession. No. MK333180.1) [19] was codon-optimized by GenSmart^TM^ Codon Optimization. The optimized p30 gene was synthesized and cloned into the plasmid pET28a using *Bam*H I and *Xho* I restriction sites in the GenScript Corporation. Then, the recombinant DNA was transformed to BL21 (DE3) chemically competent cells (TransGen, Beijing, China) and amplified at 37 °C overnight in an agar plate with kanamycin. The amplified recombinant DE3 containing recombinant DNA was verified by digestion with *Bam*H I and *Xho* I and DNA sequencing in the Sangon Biotech Corporation. The recombinant p30 with N-terminal 6 × His tag was expressed in *E. coli* (DE3) cells with 0.5 mM isopropyl-β-d-thiogalactoside (IPTG). The expression form of the recombinant p30 was checked by a sodium dodecyl sulfate-polyacrylamide gel electrophoresis (SDS-PAGE) analysis of IPTG-induced recombinant DE3 soluble and inclusion protein separated by sonication, and then recombinant p30 was purified by HisTrap HP (Cytiva, Marlborough, MA, USA). The purification procedure was conducted per the manufacturer’s guideline. Briefly, 400 mL bacterial cells induced with 1 mM IPTG were collected by centrifugation, and then resuspended with 20 mL 0.02 M PBS. Bacterial cells were disrupted using 0.2mg/mL lysozyme (Solarbio, Beijing, China) in combination with sonication. Crude lysate was centrifuged at 12,000× *g* for 15 min at 4 °C, and the pellet was resuspended in 5 mL binding buffer (8 M urea, 20 mM sodium phosphate, 0.5 M NaCl, 20 mM imidazole, pH 7.4). After centrifugation as previously described, the supernatant was collected and added to the pre-equilibrated HisTrap HP column. After washing with 15 mL binding buffer, the 6 × His tag p30 protein was eluted in fractions with 9 mL elution buffer (8 M urea, 20 mM sodium phosphate, 0.5 M NaCl, 200 mM imidazole, pH 7.4) and collected separately. The majority of purified p30 protein fractions were collected and dialyzed with 0.02 M PBS overnight at 4 °C. The purified p30 protein was analyzed by His tag rabbit polyclonal antibody (Solarbio, Beijing, China) and ASFV-positive standard serum (lot no. 202112, purchased from the China Veterinary Drug Administration).

### 2.2. Production of p30 Monoclonal Antibody

Monoclonal antibodies were produced as previously described [20]. Briefly, purified recombinant p30 was dispersed with MontanideTM Gel 01 PR (Seppic, La Garenne-Colombes, France) by gentle mixing. The final volume ratio of p30 was 80% with 500 μg p30 per 1 mL mixture, and 50 μg emulsified p30 was subcutaneous inoculated into 6-week-old BALB/c mice. Mice were immunized three times at 14-day intervals, and 3 days later from the last immunization, the mouse was euthanized to collect the splenocytes. Subsequently, SP2/0 myeloma cells were fused with splenocytes by PEG 1450 (Sigma, Saint Louis, MO, USA), and resuspended with RPMI-1640 medium containing 20% FBS and 1 × Hypoxanthine aminopterin thymidine (HAT) (Sigma, Saint Louis, MO, USA). Thereafter, the resuspension was dispensed into 96-well plates (Corning, NY, USA), and cultured under 5% CO_2_ at 37 °C. A coating antigen of p30 was used to screen the supernatants of fused cells via the indirect ELISA. Through limited dilution, confluent hybridomas in positive wells were subcloned three times to achieve a single hybrid cell. Then, amplified hybridomas (10^6^ cells) were intraperitoneally injected into BALB/c mice pretreated with liquid paraffin one week early. Nine days later, the mouse ascites were collected and centrifuged to prepare the monoclonal antibody. The prepared mAbs were separately analyzed with purified p30 protein by Western blot.

### 2.3. Indirect ELISA

Standard protocol for the indirect ELISA was conducted by the procedure described previously [18]. Briefly, purified recombinant p30 diluted in carbonate buffer was coated on 96-well plates (Corning, NY, USA) (0.5 μg/mL, 100 μL/well) and incubated overnight at 4 °C. The plate was washed three times with PBST (0.05% Tween in 0.01 M PBS) and blocked with 1% gelatin (Amresco, Pittsburgh, PA, USA) in PBST for 2 h at 37 °C, then washed three times with PBST. Supernatants (50 μL) were added and incubated for 30 min at 37 °C. As a control, p30-immunized mice and unimmunized mice sera were diluted 1:100 separately in duplicate. After incubation, the plate was washed as described beforehand. Horseradish peroxidase (HRP) conjugated goat anti-mouse IgG + IgM + IgA (Bethyl, Montgomery, TX, USA) diluted 1:20,000 in PBST was dispensed into plates at 100 μL/well, and incubated for 30 min at 37 °C. Following washing three times with PBST, 100 μL chromogenic substrate solution (TMB) (Huzhou InnoReagents Corp., Zhejiang, China) was added into each well. Incubation took place at room temperature for 10 min, followed by the addition of 50 μL of 2 M sulfuric acid per well. The result was read at OD450 nm absorbance by using a microplate reader (Bio Tek, Winooski, VT, USA).

### 2.4. Solid Phase Blocking ELISA (bELISA)

As inhibitors for the bELISA, five pig sera against ASFV were used to interfere the binding of mAb to the p30 protein, and five healthy pig sera were used as ASFV-negative serum samples. Ten antisera, as aforementioned, were selected from 138 pig serum samples verified by Western blot (Appendix A). The solid phase blocking ELISA was modified from the indirect ELISA by adding serum (diluted in PBS containing 0.5% BSA at ratio of 1:1), 100 μL/well, and negative controls consisting of duplicate wells containing pig serum (Tianhang, Zhejiang, China). Then, the plate was incubated for 60 min at 37 °C, following washing three times with PBST. Thereafter, supernatants containing mAb were added to the well and continued the indirect ELISA procedure. Measurements were conducted on the OD values with and without the serum, and calculated the percent inhibition (PI) according to the following formula:PI(%)=[(mean OD of negative control−OD of sample)/mean OD of negative control]×100(%)

### 2.5. Mapping of Epitope and Alignment of p30 Protein

Epitope mapping of the mAb was conducted by Western blot as described previously [21]. Briefly, four different fragments of DNA of the p30 gene, encoding amino acids (a.a.) 8–101; a.a. 12–158; a.a. 18–158; a.a. 58–194, were respectively cloned into plasmid pET32a. The truncated p30 proteins were expressed as a fusion protein in *E. coli* BL21 cells induced by IPTG. The induced cells were lysed in the SDS loading buffer and incubated in a boiling water bath for 10 min. The denatured proteins were then separated on 12% Bis-Tris gel, and further transferred onto nitrocellulose membranes (Millipore, Burlington, MA, USA). After blocking with 5% skimmed milk in PBST overnight at 4 °C, the membranes were washed three times with PBST. Following incubation with HRP-labeled mAb diluted 1:4000 in blocking buffer for 1 h at room temperature, the membranes were extensively washed, and the blots were scanned using a chemiluminescence image analyzer (Tanon, Shanghai, China).

The amino acid alignment of 21 p30 sequences containing 19 genotypes was conducted by software MegAlign (Appendix A). Therein, 2 sequences were, respectively, from 2 ASFV strains (genotype I and genotype II) isolated in China, and another 19 sequences containing 19 genotypes were similar to those analyzed in the previous study [20].

### 2.6. Preparation of the HRP-Labeled mAb Conjugate

The mAb ascites were purified by Protein G SefinoseTM Resin (Sangon Biotech, Shanghai, China) following the instruction. Then, the purified mAb was labeled with horseradish peroxidase (HRP) by the procedure described previously [22]. Briefly, 4 mg HRP (Sigma, Saint Louis, MO, USA) was dissolved in 0.5 mL ultrapure water, mixed with 0.5 mL NaIO4 (0.06 M), and incubated for 30 min at 4 °C. Then, 0.5 mL ethylene glycol aqueous (0.16 M) was added and incubated at room temperature for 30 min. Next, 1 mL purified mAb (4 mg/mL) was mixed with the solution mentioned above. Subsequently, the mixture was put in a dialysis bag, and slowly dialyzed in 0.05 M pH 9.5 carbonate buffer overnight at 4 °C. Thereafter, the solution in the dialysis bag was pipetted, 0.2 mL NaBH4 (5 mg/mL) was added, and it was incubated for 2 h at 4 °C. Subsequently, an equal volume of saturated ammonium sulfate was added to precipitate the labeled mAb at 4 °C for 30 min. After centrifuging at 12,000× *g*, 4 °C for 10 min, the precipitate was dissolved in 0.02 M PBS, and dialyzed with 0.02 M PBS overnight at 4 °C. The insoluble material was discarded by centrifuging again, and the supernatants were pipetted, resulting in the HRP-labeled mAb conjugate.

### 2.7. Serum Panel

The cELISA was evaluated by a panel of 138 pig serum samples (40 positive and 98 negative sera). Therein, 40 positive sera against ASFV were derived from ASFV-infected pigs, kindly gifted by the Jiangsu Academy of Agricultural Sciences Veterinary Diagnosis and Testing Center, and 98 negative sera were from healthy pigs collected before the outbreak of ASF in China by our laboratory. All samples were confirmed by Western blot with purified recombinant p30 (Appendix A).

### 2.8. Solid Phase Competitive ELISA (cELISA)

An HRP-labeled mAb based-competitive ELISA was carried out to evaluate the potential usage of mAb for ASF diagnosis as previously described [23]. Briefly, 96-well plates were coated with 0.05 μg/well recombinant p30 protein and incubated overnight at 4 °C. On the next day, the liquid in the plates was discarded and the plates were washed three times with PBST. The plates were dispensed with 1% gelatin (200 μL/well) in PBST and incubated for 2 h at 37 °C, followed by washing three times with PBST. The optimization of pig serum was beforehand conducted with serial two-fold dilution (1:2, 1:4, 1:8, 1:16, and 1:32). Each well coated with p30 antigen was incubated with 50 μL pig serum (diluted in PBS containing 0.5% BSA at ratio of 1:1) for 30 min at 37 °C. Meanwhile, the negative controls consisting of duplicate wells containing pig serum (Tianhang, Zhejiang China) were included. Later, 50 μL HRP-labeled mAb (0.03 μg, diluted 1:1600 with PBST containing 0.5% BSA) was added into the plates, followed by incubating for 30 min at 37 °C. After washing, 100 μL/well TMB was dispensed and left at room temperature for 10 min. Then, the reaction was stopped by adding 50 μL 2 M sulfuric acid per well. The result was read at OD450 nm. The raw data calculated the percent of inhibition (PI) according to the following formula:PI(%)=[(mean OD of negative control−OD of sample)/mean OD of negative control]×100(%)

### 2.9. Assessment of cELISA Specificity and Repeatability

Six polyclonal antisera against other pig viruses (CSFV, PEDV, RV, PRV, PRRSV, PCV2) were tested by the developed cELISA to analyze the specificity.

The repeatability of the cELISA was determined by analyzing eight pig sera involving three positive sera against ASFV, three moderate positive sera against ASFV, and two negative sera. The coefficient of variation (CV) was calculated according to the OD values, CV=(standard deviation SD/mean)×100%. The intra-assay CV was measured by each serum detected on three plates in one run, and the inter-assay CV was measured by each serum detected in three runs.

### 2.10. Detection Antibody in ASFV-Positive Standard Serum

The ASFV-positive standard serum (lot no. 202112) was serially two-fold diluted from 1:4 to 1:1024, then the PI value was tested by the established cELISA. The highest dilution of the serum in which the PI value exceeded the cutoff value was the titer of the ASFV-positive standard serum. In addition, the titer of the ASFV-positive standard serum was determined by a commercial ASFV antibody detection kit (ID Screen^®^ African Swine Fever Competition, ID-vet, Grabels, France).

### 2.11. Data Analysis

To confirm the cutoff value, and corresponding diagnostic sensitivity and specificity, a panel of serum samples was tested by cELISA. A receiver operating characteristic (ROC) analysis was conducted using GraphPad Prism 9.4.0. By the validated serum samples tested by Western blot, the ROC analysis would automatically determine the cutoff value which yielded optimized sensitivity and specificity. The coefficient of variation (CV) was calculated using SPSS software for Windows, version 17.0.

## 3. Results

### 3.1. Expression of Recombinant p30 Protein

The codon-optimized CP204L gene originated from ASFV Pig/HLJ/2018 was synthesized and expressed in *E. coli* BL21 as an N-terminal His-tagged recombinant protein. The p30 protein was expressed as inclusion bodies, and it reached high purity after purification by HisTrap HP (Cytiva, Marlborough, MA, USA). The approximate 33kDa fusion protein of the recombinant p30 was observed by sodium dodecyl sulfate-polyacrylamide gel electrophoresis (SDS-PAGE) (Figure 1A) and confirmed by Western blot using a His tag rabbit polyclonal antibody and the ASFV-positive standard serum (Figure 1B).

### 3.2. Production of mAbs against p30 of ASFV

To generate mAbs against the p30 of ASFV, BALB/c mice were immunized with recombinant p30 protein. The splenocytes from immunized mice were fused with SP2/0 myeloma cells, and four positive clones designated as 1D11, 1F2, 2B4, and 2H11 were achieved and subcloned by a limited dilution. Furthermore, isotypes of mAbs were analyzed using the mouse Ig isotyping kit (Biodragon, Beijing, China), and all mAbs were found to be IgG1 with the kappa light chain (Table 1). Four mAbs reacted with 33 kDa recombinant p30 by Western blot, and mAb-2B4 showed the strongest immunoreactivity (Figure 1C).

### 3.3. Assessing Potential of mAbs as Reagent in bELISA

Blocking ELISA was carried out to analyze the potential usage of these mAbs as a diagnostic reagent for ASFV antibody detection. Ten antisera were selected from 138 pig serum samples verified by Western blot (Appendix A). The bELISA result demonstrated that all five positive sera against ASFV could block mAb-2B4 by greater than 60% with a mean PI value of 70%, and a PI value less than 20% was determined for five negative sera (Figure 2). In addition, mAb-2B4 showed good reactivity with the p30 protein (Figure 1C). Therefore, monoclonal antibody 2B4 was more suitable to be inhibited by positive sera in ASFV antibody detection.

### 3.4. Preparation of the HRP-Labeled mAb-2B4 and Its Antigenic Epitope Screening

The purified mAb-2B4 was denatured to one 50-kDa and one 28-kD band in SDS-PAGE, and HRP-labeled mAb-2B4 yielded the additional 40-kD band of HRP and other bands above 75-kD (Figure 3A). By A280 nm absorption, the concentrations of the purified mAb-2B4 and HRP-labeled were determined as 4.0 mg/mL and 1.0 mg/mL, respectively. Five different fragments of p30 gene were expressed in *E. coli*, and then used to screen the epitope recognized by HRP-labeled mAb-2B4 (Figure 3B–D). The result deduced that mAb-2B4 could recognize aa 12–18 (EVIFKTD) of p30. Alignment revealed this epitope was conserved between ASFV genotype I (UEN73102.1) and genotype II (QBH90581.1), i.e., two ASFV strains isolated at different times in China. In addition, it was also highly conserved between 19 other p30 protein sequences containing 19 ASFV genotypes (Appendix A). A serological method based on conserved epitopes may provide broader diagnostic coverage.

### 3.5. Establishment of cELISA Based on HRP-Labeled mAb-2B4

In order to optimize the serum sample dilution, four known ASFV-positive and four known negative serum samples were chosen to analyze their reactiveness to a fixed dilution of HRP-labeled mAb-2B4 (0.03 μg/well), and the higher PI values of ASFV-positive sera were determined at dilution 1:2. In contrast, the PI values of negative sera were below 10% (Figure 4). Therefore, an optimal inhibition serum dilution at 1:2 with HRP-labeled mAb 2B4 (0.03 μg/well) was conducted throughout the experiment.

Following optimization of the cELISA, a panel of 138 pig serum samples (40 positive and 98 negative sera) were tested, and the PI values of each sample were calculated. A ROC analysis was conducted, and the cutoff value was determined under appropriate diagnostic sensitivity and specificity of the assay (Figure 5A). An interactive dot plot diagram showed the competitive effect of the serum samples (Figure 5B). In total, 2 out of 98 negative sera were determined false positive results with PI values of 39.4% and 43.6%, while 1 out of 40 positive sera was determined as false negative, with a PI value of 29%. The false negative serum was corresponded to lane 37 sample analyzed by Western blot, which showed a weak reactivity with p30 protein (Appendix A). The area under the curve (AUC) of the established cELISA was 0.998 (95% confidence interval: 0.994 to 1.0). In addition, a diagnostic sensitivity of 97.5% (95% confidence interval: 87.12% to 99.87%) and a specificity of 97.96% (95% confidence interval: 92.86% to 99.64%) were achieved when the cutoff value was set to 37.1%, which showed the good accuracy of the assay.

### 3.6. Assessment of cELISA Specificity and Repeatability

To assess the specificity of the established cELISA, six pig sera against other viruses (CSFV, PEDV, RV, PRV, PRRSV, PCV2) were tested. All of the non-specific positive serums were determined as the ASFV-negative serum with PI values definitely less than the cutoff value (Figure 6). The PI value for ASFV-positive serum was approximate 90.3%, while the PI values for non-specific positive serums ranged from −0.3% to 11.57%, showing good analytical specificity of the developed cELISA.

Repeatability is an important index of reliability and determines the consistency of an experiment. In this study, eight serum samples were analyzed by the established cELISA, and the coefficient of variation (CV) of each serum’s OD values was measured to assess the intra- and inter-assay repeatability. The result showed an intra-assay CV within 2.4–6.0% and an inter-assay CV within 1.3–8.3% (Table 2). The CVs of intra- and inter-assay were all below 10%, which demonstrated an adequate repeatability of the established cELISA [18].

### 3.7. Analytical Sensitivity of the cELISA

Following the optimization of the cELISA, the analytical sensitivity was assessed by the ASFV-positive standard serum. The titer of the ASFV-positive standard serum detected at different dilutions was 1:256, and the actual addition of the diluted sera was 25 μL. Meanwhile, the titer of the ASFV-positive standard serum was determined at 1:512 when using a commercial ASFV antibody detection kit, and the actual addition of the diluted sera was 50 μL according to the instruction. Therefore, the cELISA had an approximate sensitivity to the commercial kit (Figure 7).

## 4. Discussion

Although an extensive effort has been made to understand the ASFV, there is no safe and effective vaccine available to prevent ASF. The effective control of ASFV primarily depends on its early accurate diagnosis, strict movement control, and other biosecurity measurements [24]. Detection of the antibody used in the judgement of ASFV infection seems to be more crucial in the appearance of attenuated ASFV and atypical clinical symptoms. Several antibody detection ELISAs can be commercially available; however, their sensitivity needs to be further improved when compared with the confirmatory IPT [25]. The expression of p30 is observed from 2 to 4 h post-infection (hpi) of macrophages, and p30 presents in the cytoplasm throughout the infection cycle [26]. In this regard, p30 is a good candidate for the early detection of ASF. The antigenic characterization of p30 will help to improve the p30-based serological methods for the diagnosis of ASF. In the present study, we developed four murine hybridomas producing mAbs against ASFV p30, and isotypes of four mAbs were all determined to be IgG1 with the kappa light chain. Epitope mapping showed mAb-2B4 could recognize aa 12–18 (EVIFKTD) located in the N-terminal of p30, which was identical to the newly found epitope recognized by mAb 6H9A10 [21]. The biotinylated mAb 6H9A10-colloidal gold conjugate showed good ability to capture the p30 released from infected cells, and the sensitivity of the colloidal gold test strip was 2.16 ng of p30 [21]. Therefore, we speculated the high affinity to p30 of the mAb-2B4. In our study, the bELISA revealed mAb-2B4 had a good ability to discriminate the ASFV-positive serum from the ASFV-negative serum.

The amino acid residues 111–130 of p30 have been revealed as an immunodominant region [27]. Wu et al. defined 4 antigenic regions with 14 out of 21 monoclonal antibodies against ASFV p30, and region 3 and region 4 were fixed in the C-terminal of p30. In addition, the PI index indicated that region 3 and region 4 were immunodominant in response to the humoral immunity of the host. However, five overlapping p30 fragments used to screen the epitopes of the 21 mAbs were from 61aa to 201 aa of p30, omitting the epitope analysis of N-terminal of p30. While analyzing six pig sera against ASFV originated differently, the highest PI in the solid phase blocking ELISA was approximately 75% [28]. In the present study, we found the mAb-2B4 recognized the epitope of aa 12–18 of p30, which showed the average of a 70% PI value in the bELISA. We hypothesized that mAb-2B4 may recognize an immunodominant epitope on the p30 protein, and mAb-2B4 had an excellent competitive effect with the p30 antibody in the ASFV-positive serum. Since the isolation of the ASFV genotype I in China, multiple studies have developed various multiplex PCRs which can simultaneously detect genotype I and II ASFVs [29,30,31,32,33]. However, there were few cases reported of ASFV genotype I strains when the aforementioned PCRs were introduced to analyze the filed samples [30,31,33]. Animal challenge testing proved the efficient transmissibility of genotype I ASFV strain SD/DY-1/21 in pigs, and infected pigs developed low-level viremia, which makes early diagnosis more difficult than attenuated genotype II strains in the field. This may be a possible reason for illustrating the low detection rate of genotype I ASFV in clinical samples. Therefore, it is necessary to explore the supplementary assay definitely covering different genotype ASFV strains. The mAb is the vital ingredient in developing a cELISA assay, and the conserved epitope recognized by mAb will broaden the application scope. It is worth noting that aa 12–18 (EVIFKTD) was highly conserved between ASFV genotype I (UEN73102.1) and genotype II (QBH90581.1). Theoretically, the established cELISA could be used to detect the serum in the pig infected by ASFV genotype I or genotype II strains. This may facilitate efficient prevention and control for ASFV in China. Furthermore, the p30 protein sequence comparison of genotype I to genotype XVI and XIX to XXI, revealed that aa 12–18 was conserved among these 19 ASFV genotypes. A serological method based on conserved epitopes may provide broader diagnostic coverage, suggesting that the mAb 2B4 should be able to detect antibodies against 19 genotypes of ASFV.

The panels of true negative serum and true positive serum are very important to the validation of a newly established assay. We chose 98 negative pig sera collected before the outbreak of ASF in China, and 40 positive sera from pigs infected with ASFV to construct the sera panel. Additionally, these sera were confirmed by using a commercial ASFV antibody detection kit. The HRP-labeled mAb-2B4-based cELISA showed the optimal balance of specificity of 97.96% and sensitivity of 97.5%. In the previous study, Yu et al. established a p30 mAb-based bELISA with the specificity of 98.96% and sensitivity of 97.96%, and they also detected seroconversion in two out of five pigs at 10 days post-infection with ASFV [18]. Subject to the absence of a level 3 biosafety laboratory, we could not process that analysis by cELISA. Consequently, we assessed the sensitivity of cELISA by calculating the titer of the standard ASFV-positive serum. The result demonstrated the titer of the ASFV-positive standard serum with serially two-fold dilutions was 1:256, while its titer was 1:512 in the previously established p30 mAb-based bELISA [18]. Although the lot no. of the standard serum used in the two assays were different, we speculated that the reason for the decline of the titer might be the different manner of incubation between the blocking ELISA and cELISA. In the blocking ELISA, the sample serum was discarded after incubation, while the sample serum was incubated coupled with the HRP-labeled mAb in the cELISA. Ingredients in the serum, such as autoantibodies, hemolysis, and lipemia can affect the serologic testing [34,35].

The clinical symptoms of ASF are difficult to discriminate from PRRS, CSF, PMWS, and SE [36]. The cELISA developed demonstrated good specificity and no cross-reactivity with six pig sera against other viruses (CSFV, PEDV, RV, PRV, PRRSV, PCV2). An intra- and inter-assay also revealed excellent repeatability. Overall, we developed a specific, sensitive, and low-cost cELISA on an HRP-labeled mAb with a conserved epitope.

## Figures and Tables

**Figure 1 viruses-15-00154-f001:**
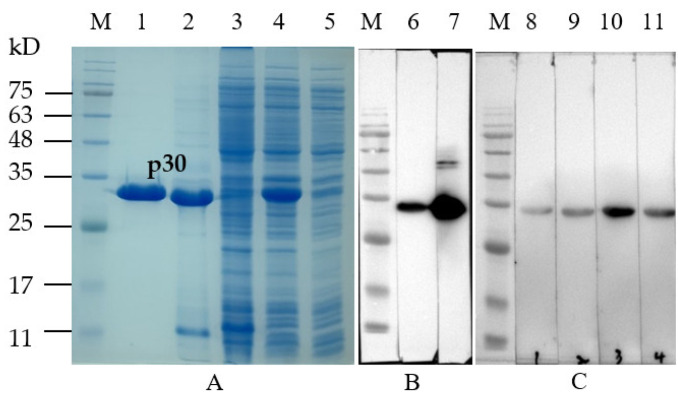
Analysis of p30 protein and mAbs: (**A**) SDS-PAGE analysis of p30 protein; M–protein marker, 1—purified p30, 2—pellet of bacterial cell lysates induced by 0.5 M IPTG, 3—supernatants of bacterial cell lysates induced by 0.5M IPTG, 4—whole-cell lysates of induced bacterial, 5—whole-cell lysates of uninduced bacterial. (**B**) Western blot analysis of the purified p30 protein with anti-His tag rabbit polyclonal antibody and ASFV-positive standard serum; 6—His tag rabbit polyclonal antibody, 7—ASFV-positive standard serum. (**C**) Western blot analysis of reactivity of mAbs with the purified p30; 8—mAb-1F2, 9—mAb-1D11, 10—mAb-2B4, 11—mAb-2H11.

**Figure 2 viruses-15-00154-f002:**
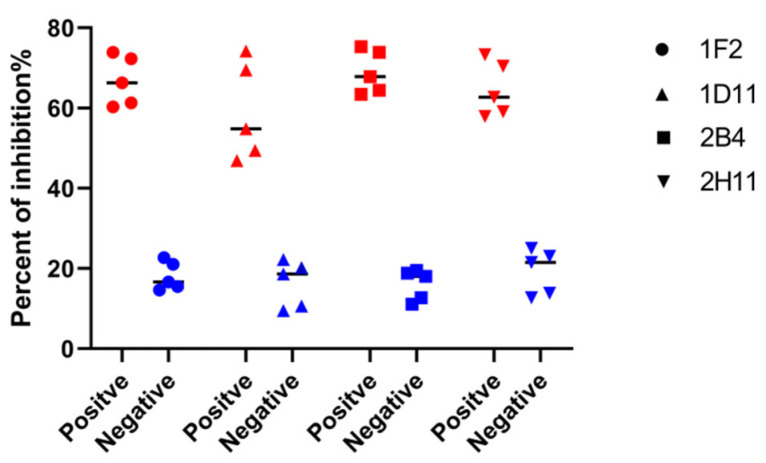
Analysis of mAbs on iELISA for ASFV antibody detection. Each symbol shape represents sera inhibition of identity of mAb, respectively. Five positive sera against ASFV (Red) and five negative sera (Blue) were dotted, and the average percent of inhibition of positive and negative sera was recorded in each mAb.

**Figure 3 viruses-15-00154-f003:**
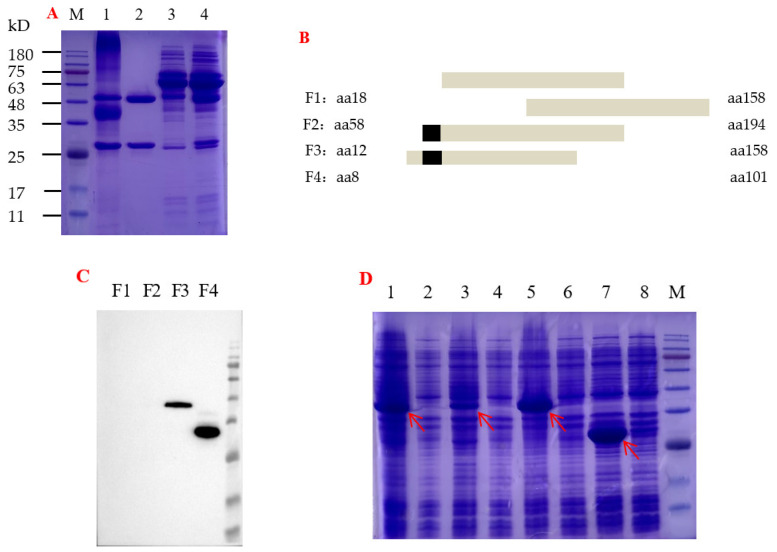
Purification and epitope screening of mAb-2B4: (**A**) SDS-PAGE analysis of purified mAb-2B4; M–protein marker; 1—HRP-labeled mAb-2B4; 2—purified mAb-2B4, diluted 1:10; 3—flow-through pool, diluted 1:10; 4—mouse ascites, diluted 1:25. (**B**) The sketch map of truncated fragments of p30 expressed in *E. coli* prepared for epitope screening. The black box was deduced to the epitope recognized by mAb-2B4. (**C**) Epitope (gray box and black box in (**B**)) of p30 recognized by HRP-labeled mAb-2B4 using Western blot. (**D**) SDS-PAGE analysis of four truncated p30 expressed in *E. coli*. Red arrows show four truncated recombinant p30 protein; M–protein marker; 1, 3, 5, 7—corresponded to whole-cell lysates of induced F1, F2, F3, F4, respectively; 2, 4, 6, 8—corresponded to whole-cell lysates of pre-induced F1, F2, F3, F4, respectively.

**Figure 4 viruses-15-00154-f004:**
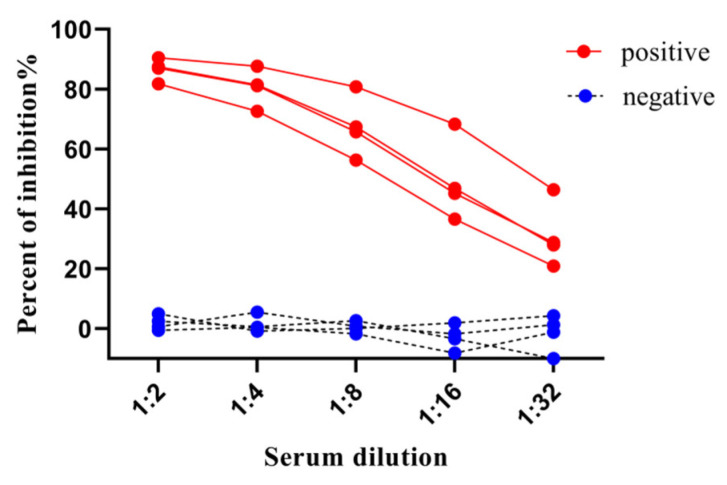
Optimization of serum dilution in cELISA. Serial two-fold dilution of four positive and four negative serum samples were tested in cELISA and serum dilution of 1:2 displayed higher percent of inhibition in four positive serum samples.

**Figure 5 viruses-15-00154-f005:**
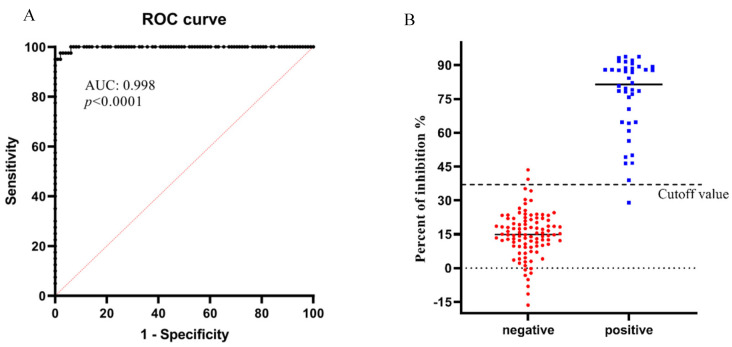
Receiver operating characteristic (ROC) analysis for ASFV p30-based cELISA. The assay was conducted by ASFV-negative sera (n = 98) and ASFV-positive sera (n = 40): (**A**) ROC analysis of cELISA result with the area under the curve (AUC) of the test was 0.998. (**B**) Interactive dot plot diagram displaying the PI values of sera while the cutoff value was set to 37.1%.

**Figure 6 viruses-15-00154-f006:**
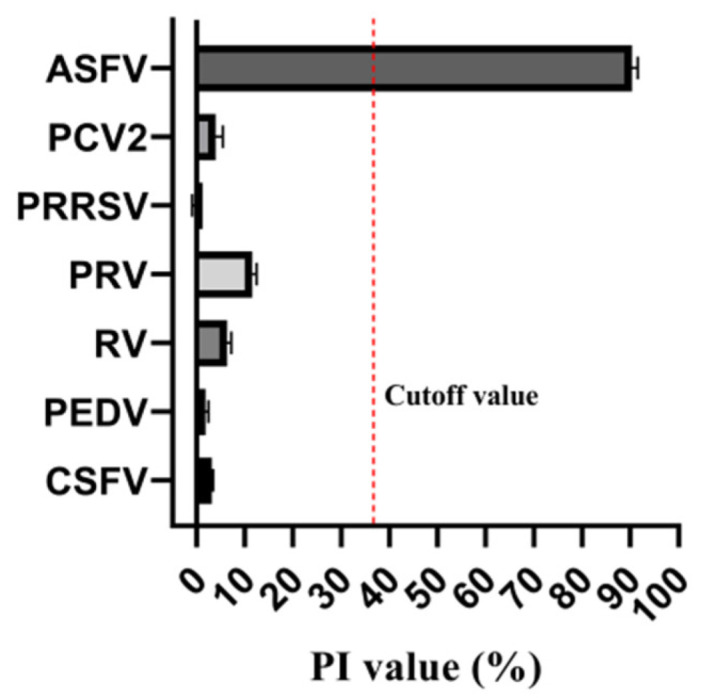
Percent of inhibition of values of various porcine viruses after detection of cELISA. All non-specific positive serums were classified as ASFV-negative serums, while the PI value of the ASFV-positive serum exceeding the cutoff value.

**Figure 7 viruses-15-00154-f007:**
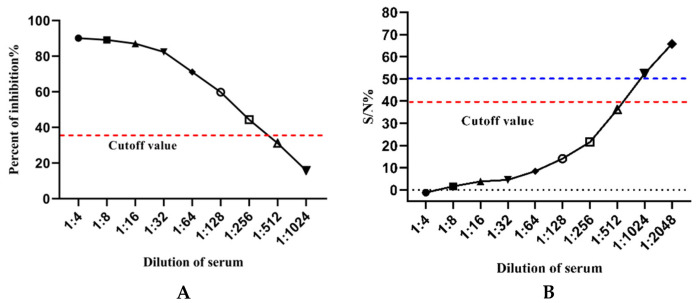
Sensitivity assay of different ELISAs. The assay was conducted by testing the titer of ASFV-positive standard serum: (**A**) The established cELISA. Positive serums diluted less than or equal to 1:256 were all determined positive for higher PI values than the cutoff value. (**B**) Commercial ASFV antibody detection kit. $\mathrm{SN}\left(\%\right)=\left[\left(\mathrm{OD}\;\mathrm{of}\;\mathrm{sample}-\mathrm{mean}\;\mathrm{OD}\;\mathrm{of}\;\mathrm{positive}\;\mathrm{control}\right)/\left(\begin{array}{c} \mathrm{mean}\;\mathrm{OD}\;\mathrm{of}\;\mathrm{negative}\;\mathrm{control}-\\ \mathrm{mean}\;\mathrm{OD}\;\mathrm{of}\;\mathrm{positive}\;\mathrm{control} \end{array}\right)\right]\times 100\left(\%\right)$. According to the instruction, the sample tested with SN value less than or equal to 40% was determined to be ASFV positive, while SN value more than or equal to 50% was determined to ASFV negative. Positive serums diluted less than or equal to 1:512 were all determined positive for lower SN values than the cutoff value.

**Table 1 viruses-15-00154-t001:** Identification of isotypes of p30 monoclonal antibodies.

	Monoclonal Antibodies			
	1F2	1D11	2B4	2H11
Ig subclass	IgG1	IgG1	IgG1	IgG1
Light chain type	κ	κ	κ	κ

**Table 2 viruses-15-00154-t002:** Repeatability of the established cELISA.

Sera	Intra-Assay			Inter-Assay		
	Mean OD	SD	CV%	Mean OD	SD	CV%
1	0.145	0.005	3.4	0.145	0.012	8.1
2	0.172	0.010	6.0	0.167	0.013	7.7
3	0.187	0.005	2.4	0.206	0.017	8.3
4	0.574	0.019	3.3	0.551	0.037	6.7
5	0.528	0.023	4.4	0.579	0.025	4.3
6	0.513	0.020	3.8	0.537	0.021	3.8
7	0.997	0.028	2.9	0.940	0.012	1.3
8	1.033	0.035	3.4	0.992	0.023	2.3

## Data Availability

All data used and presented in this study are either available in public repositories as described in the Section 2, or are made available in NCBI database.

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
