# Peer review of "Development of a Competitive Enzyme-Linked Immunosorbent Assay Targeting the-p30 Protein for Detection of Antibodies against African Swine Fever Virus"

_viruses, 2023, doi:10.3390/v15010154_

Round 1
Reviewer 1 Report (Previous Reviewer 2)
The revised version is highly improved, and the importance of the newly developed p30-based ELISA is clear now. However, a number of errors or concerns must further be addressed and elaborated.
Major comments:
1. Reference is missing on line 43-46.
2. Supplementary figures should be quoted on line 126, 166.
3. Line 187: What is ‘identity serum’?
4. Line 235-236: The meaning of two sentences is contradict. The ‘well reactivity with p30 protein’ should indicate high affinity with p30 protein. If this is the case, why is antibody 2B4 more suitable to be inhibited by ASFV positive sera?
5. Line 248: Why are there two additional bands from HRP-labeled 2B4? What are they?
6. Line252-253: It is recommended to highlight the sequence on Supple Fig 2 and to quote the sentence with the corresponding Fig.
7. Line272: What is mouse ascites? This protein was not mentioned in M&M.
8. Line279: The sub-title should be rephased.
9. What does the red line indicate in Fig 5, and should the x-axis be labeled as ‘specificity’ instead of ‘1-specificity’? It is recommended to include the following sentence as found in the responses to reviewers: “This ROC analysis would automatically determine the cutoff value which yielded optimized sensitivity and specificity.”
10. The title of Supple Fig 1 should be included, for example, ‘Specificity test of ASFV positive and negative sera with purified recombinant p30 (0.4 ug/lane)’.
11. Are there any reference CV values to determine the repeatability of ELISA? Why does the CV value below 10% demonstrate the ‘adequate’ repeatability?
12. Line 342: How is the good sensitivity of the ELISA defined? The results only revealed that the established cELISA had similar sensitivity to the commercial one.
13. In Fig7B, “S/N %” is not defined. It is hard to compare the results between established cELISA and the commercial detection kit.
14. Line 368-370: The following idea should be emphasized/included in the discussion: ‘Due to the identity of the epitope recognized by mAb 6H9A10 and mAb-2B4, we wanted to emphasis the high affinity to p30 of the mAb-2B.’
15. Line 402: It is highly recommended to specify ‘that analysis’ as found in the responses to reviewers.
Minor comments:
a. There are still some grammatic/typo- errors, line 50, 54, 88, 152, 154, 358-359
b. ‘p30’ has to be unified throughout the whole manuscript.
Author Response
Please see the attachment.

Reviewer 2 Report (New Reviewer)
The document was greatly improved reaching a suitable form to be accepted for publication.
Author Response
Please see the attachment.

Reviewer 3 Report (New Reviewer)
1. Line 163,40 positive sera were from infected pigs. The description is not clear and precise. The sera should be indicated that they were derived from ASFV-infected pigs, in addition, positive sera should be mentioned clearly that they were antisera against ASFV.
2. Recombinant p30-based Western blot is not a gold standard assay for ASFV, they should confirm the serum samples by immunofluorescence assay with ASFV-infected cells.
Author Response
Please see the attachment.

This manuscript is a resubmission of an earlier submission. The following is a list of the peer review reports and author responses from that submission.
Round 1
Reviewer 1 Report
1. The English writing need to be polished carefully. For example, The sentence in line 43-45 “The severity of the diseases largely depends on the virus virulent, incubation period in nature is usually 4-19 days, and acute form 3-4 days” should be “The severity of the diseases largely depends on the virus virulence. It’s incubation period in nature is usually 4-19 days, and acute form 3-4 days.” And also, the abbreviation OIE in line 55 should be WOAH now.
2. In line 113-115. The infection background of the 5 positive and five negative sera were determined by the commercial ID-vet Kit. Which should be identified with gold standard such as IFA assay. At the same time, the field samples used as serum panel should also be tested with gold standard assays.
Reviewer 2 Report
African swine fever virus (ASFV) has been widely spread in Africa, Asia and Europe since 1921. From 2018, there has been an outbreak in China, causing hundreds of millions of pigs’ deaths. Until now, no vaccines and drugs are available. It is important to have a rapid and efficient method to control and monitor the spread. In this study, Zhou and colleagues isolated and purified p30 monoclonal antibody (mAb) for the development of ELISA, which could finally help the virus surveillance. The purified p30 mAb is useful and important, and the epitope mapping data is informative. However, the significance of the established ELISA, named competitive ELISA (cELISA), is questionable. One of the major reasons is that the ELISA setup, procedure, and components are quite comparable to the currently available blocking ELISA (bELISA). In addition, number of major concerns must be addressed.
Major comments:
1. In some sentences, proper references are missing, especially in introduction and discussion sections. Line 36, 38, 327.
2. Line 50-51, the idea of two sentences was not cohesive.
3. Purification procedure of p30 is missing in the M&M section. In Fig 1, most of the p30 was found in the pellet. It is critical to describe the whole purification procedure. The purification of p30 is questionable.
4. In line 116, what component in the indirect ELISA is replaced by serum?
5. In some sentences, the wording of ‘pig’ is used while ‘swine’ is used in others. It is highly recommended to unify the wording.
6. Line 171, the coefficient variation (CV) is not defined. How are the CV values obtained or calculated?
7. In Fig 1 B &C, what samples were used for the Western?
8. Fig 2 showed that 1F2 has comparable percent of inhibition to 2B4, which was not clearly described. Why 2B4 is specifically selected as the hit for further usage? Further discussion is required.
9. It is highly recommended to include the purified HRP-2B4 mAb sample in Fig 3A. In addition, Fig 3D should be represented before Fig 3B&C. In addition, the corresponding non-induced samples in Fig 3C has to be presented.
10. Line232, ‘this epitope was highly conserved between ASFV genotype I and genotype II’. The sequence alignment is required.
11. In Fig 5A, y-axis label is missing.
12. The calculation and the meaning of ROC curve was not clearly described. The axis label of Fig 5B has some problems.
13. How is the CV value of the intra- and inter-assay calculated? Corresponding results have to be presented.
14. Line 306, how is a good ELISA assay defined? Which assay is used as the reference?
15. Line 327-327, what purpose is it to mention the biotinylated mAb 6H9A10-related assay? Is there any comparisons between the established cELISA or any commercially available ELISA?
16. Line 341-342, it is recommended to have the sequence alignment of different ASFV genotype.
17. Line 351-352, which analysis is required to perform in BSL3 lab?
18. The significance of the established cELISA is not clearly discussed. First, the established one seems not to be as good as the currently available bELISA (line 352-359). Second, the cost of this established cELISA is questionable (line 363). In-depth discussion is required to support the importance and usage of the cELISA. The detection of other genotypes with this cELISA is recommended.
19. There are many typo- or grammatical errors, especially in the M&M section. It is highly recommended to have more clear and precise description of all the methods. Careful proof-reading must be required: Line 16, 39, 42-46, 52-54, 62-63, 72, 84, 92, 94, 100-102, 107, 113, 127, 135, 137-140, 142, 155-157, 162, 177, 205, 224, 230, 319, 321, 323, 327, 332-334, 357, 360.